# Diversity and Depth in Per-Example Routing Models

**Prajit Ramachandran**
Google Brain
`prajit@google.com`

**Quoc V. Le**
Google Brain
`qvl@google.com`

## Abstract

Routing models, a form of conditional computation where examples are routed through a subset of components in a larger network, have shown promising results in recent works. Surprisingly, routing models to date have lacked important properties, such as architectural diversity and large numbers of routing decisions. Both architectural diversity and routing depth can increase the representational power of a routing network. In this work, we address both of these deficiencies. We discuss the significance of architectural diversity in routing models, and explain the tradeoffs between capacity and optimization when increasing routing depth. In our experiments, we find that adding architectural diversity to routing models significantly improves performance, cutting the error rates of a strong baseline by 35% on an Omniglot setup. However, when scaling up routing depth, we find that modern routing techniques struggle with optimization. We conclude by discussing both the positive and negative results, and suggest directions for future research.

## 1 Introduction

Modern neural networks process each input in the exact same way. This static paradigm is rigid compared to how brains process sensory inputs. Brains can utilize different subnetworks to process different categories of objects, such as face-specific processing in the fusiform face area (Kanwisher et al., 1997; Grill-Spector & Weiner, 2014). While static neural networks are empirically effective, it remains an open question whether neural networks with input-dependent processing can improve performance. Input-dependent processing holds the promise of offering better parameter efficiency and reduced computation due to the specialization of processing.

Input-dependent processing has been underexplored in comparison with the wealth of work on static networks. Much of the work exploring input-dependent processing has taken the form of per-example routing within a network (Shazeer et al., 2017; Fernando et al., 2017; McGill & Perona, 2017; Rosenbaum et al., 2018), which is form of conditional computation (Bengio et al., 2013). In per-example routing, different examples are processed by different subcomponents, or *experts* (Jacobs et al., 1991), inside a larger model, or *supernetwork* (Fernando et al., 2017). Only a subset of experts in the supernetwork are active for any given example. This paradigm enables the experts, each of which has its own set of parameters, to specialize to subsets of the input domain. The process of *routing* each example, which determines the experts that are used, is learned jointly with the parameters of the experts.

Routing models to date have been relatively small, homogeneous networks. Typically, the same architectural unit (e.g., a fully connected layer of the same width) is used for every expert. The experts differ only in the parameters. Intuitively, the diversity of input examples is best handled by a diversity of architectural units with varying properties, implying that the usage of homogeneous experts is limiting. Furthermore, the number of routing decisions made in prior routing network works has typically been five or fewer. More routing decisions increase the number of distinct paths in the network, which may increase representational power. Making static networks deeper reliably improves performance, so we suspect that the representational power of routing networks is limited when only a few routing decisions are made.

In this work, we address these two deficiencies in routing models. Since we aim to increase the representational capacities of routing models, we first introduce a simple trick that reduces overfitting.

We then show how routing models with architectural diversity represent a broad family of models that generalize a number of powerful models. We also discuss the tradeoffs of scaling up the number of routing decisions with respect to optimization difficulty. In our experiments, we demonstrate that architecturally diverse routing models beat the best baselines with a 35% improvement in error rate on an Omniglot setup. By ablating the architectural diversity, we show that diversity plays a key role in achieving strong performance. We then scale up the number of decisions in routing models on CIFAR-10 and demonstrate that while competitive performance can be achieved, the accuracy drops as the number of decisions increase due to optimization challenges. Finally, we discuss our both our positive and negative findings and suggest future research directions for routing models.

## 2 METHODS

### 2.1 ROUTING

Static neural network architectures apply the same function to every example. In contrast, input-dependent models attempt to tailor the function to each example. While it is straightforward for a human to manually specify a single static architecture, it is infeasible to specify every input-dependent function by hand. Instead, the input-dependent function must be automatically inferred by the model, which introduces an extra level of complexity in optimization.

Given the need to automatically infer architectures for each example, a natural solution is to define a single large model (*supernetwork*) with a numerous subnetworks (*experts*), and *route* examples through a path in the supernetwork (Fernando et al., 2017; Rosenbaum et al., 2018; Pham et al., 2018; Bender et al., 2018). Figure 1 visualizes this framework. Intuitively, similar examples can be routed through similar paths and dissimilar examples can be routed through different paths. The example-dependent routing also encourages expert specialization, in which experts devote their representational capacity to transforming a chosen subset of examples.

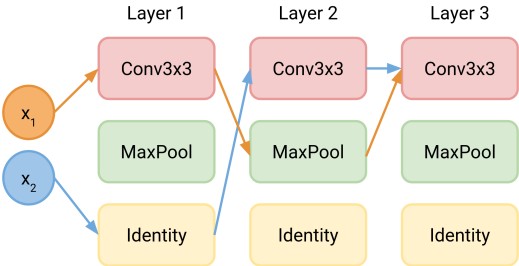

Figure 1: An example of a routing network.

Learning to route examples to well-matched experts is critical for good performance. Effective routing can be achieved by training another small neural network (*router*) that learns to route examples through the supernetwork. The router takes the example as input and outputs the next expert to use. The router can take advantage of the intermediate representations of the example produced in the supernetwork.

Given that the router is characterized as neural network, it must be trained. One option is to use reinforcement learning (Rosenbaum et al., 2018). In this work, we use the noisy top-$k$ gating technique of Shazeer et al. (2017) that enables the learning of the router directly by gradient descent. Noisy top-$k$ gating is used to choose $k$ experts out of a list of $n$ experts in a way that enables gradients

to flow to the router. The noisy top-$k$ gating technique is defined as:

$$y = \sum_i^n G(x)_i E_i(x)$$

$$G(x) = \texttt{softmax}(\texttt{keep-top-k}(H(x), k))$$

$$H(x)_i = (x \cdot W_g)_i + \epsilon \cdot \log\left(1 + \exp\left((x \cdot W_{\texttt{noise}})_i\right)\right)$$

$$\texttt{keep-top-k}(v, k)_i = \begin{cases} v_i & \text{if } v_i \text{ is in the top } k \text{ elements of } v \\ -\infty & \text{otherwise} \end{cases}$$

$$\epsilon \sim \mathcal{N}(0, 1)$$

where $y$ is the output, $G(x)$ are the expert weights, and $E(x)_i$ is the $i^{\text{th}}$ expert. Both $W_g$ and $W_{\texttt{noise}}$ can be learned using the gradients that flow through $G(x)$. While the `keep-top-k` operation introduces discontinuities in the output, gradient descent learning with noisy top-$k$ gating works well in practice. The induced sparsity of choosing $k$ components significantly reduces computational cost, in contrast to soft gating mechanisms where the entire supernetwork is activated (Eigen et al., 2013).

The number of routes taken by the examples is controlled by the hyperparameter $k$. The choice of $k$ plays an important role in controlling the tradeoff between weight sharing and specialization. We find that using a small fixed $k$ throughout the network leads to overfitting. To combat overfitting, we introduce a simple trick which we call *k-annealing*. In $k$-annealing, instead of using the same value of $k$ for every layer, $k$ is annealed downwards over the layers of the routing model. That is, initial layers will have high $k$ (more weight sharing) and layers closer to the classification layer have smaller $k$ (more specialization). We found in early experimentation that $k$-annealing works well in practice, and we use it for all experiments in this work.

If noisy top-$k$ gating is used without modifications, the model tends to immediately collapse to using a small number of experts instead of using all the capacity. The collapse happens in the beginning of training. Some experts tend to perform slightly better due to random initialization, and gradient descent will accordingly route more examples to the best few experts, further reinforcing their dominance. To combat this problem, Shazeer et al. (2017) used additional losses that balance expert utilization, such as the importance loss:

$$\mathcal{L}_{\texttt{importance}}(X) = \texttt{cv}\left(\sum_{x \in X} G(x)\right)^2$$

where `cv` is the coefficient of variation. We use the importance loss and the load loss, which is defined in detail in Shazeer et al. (2017), in all our models.

## 2.2 DIVERSITY

Previous works on routing use a homogeneous set of experts, where all experts share the same architecture and only differ in the parameter values (Shazeer et al., 2017; Rosenbaum et al., 2018). Intuitively, this pattern is suboptimal because different examples may need to be processed by architectural units with different properties. For example, Figure 2 shows two images from Imagenet (Russakovsky et al., 2015) of borzoi dogs from the same class. The borzoi on the left fills the image and may require layers that can combine information across the entirety of the image, such as convolutions with large spatial extent, whereas the borzoi on the right is less prominent and may require layers, such as max pooling, that can ignore the distracting background information.

We attempt to remedy this problem by introducing architectural diversity in routing models, with different experts having different architectures and parameters. The simplest example of this pattern is a feedforward network with multiple layers. In a standard static neural network, an example would be processed as:

$$h_i^{l+1} = \phi(\texttt{conv-3x3}(h_i^l))$$

where $\phi$ denotes a nonlinearity. In contrast, a model with diverse routing could be specified as:

$$h_i^{l+1} = \texttt{gating}_k([h_i^l, \ \phi(\texttt{conv-3x3}(h_i^l)), \ \phi(\texttt{conv-7x7}(h_i^l)), \ \texttt{max-pool-3x3}(h_i^l)])$$

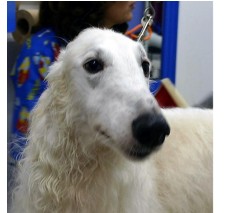 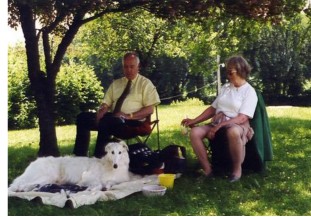

Figure 2: Two Imagenet images from the same class (borzoi, synset *n02090622*).

Even this simple structure can represent a number of computational models. If the example requires gathering information from a large receptive field, the `conv-7x7` can be used. If skip connections (He et al., 2016) are beneficial, the router can choose the identity operation.

Interestingly, introducing architectural diversity into routing models can be viewed as a way to generalize a number of recently proposed models. One-shot neural architecture search models (Brock et al., 2017; Pham et al., 2018; Bender et al., 2018) jointly perform architecture learning and parameter learning in the same supernetwork. These one-shot models are just a special case of diverse routing models when the input to the router is a constant across examples, so every example follows the same path inside the supernetwork. Diverse routing models can be extended to the multi-task setting by simply concatenating a task embedding to the input of the routing model (Johnson et al., 2016), with the task embedding being shared across all examples of the same task. If only the task embedding is fed as input to the router, then all examples from the same task will be routed through the same path, just as in task-based routing models such as Pathnet (Fernando et al., 2017).

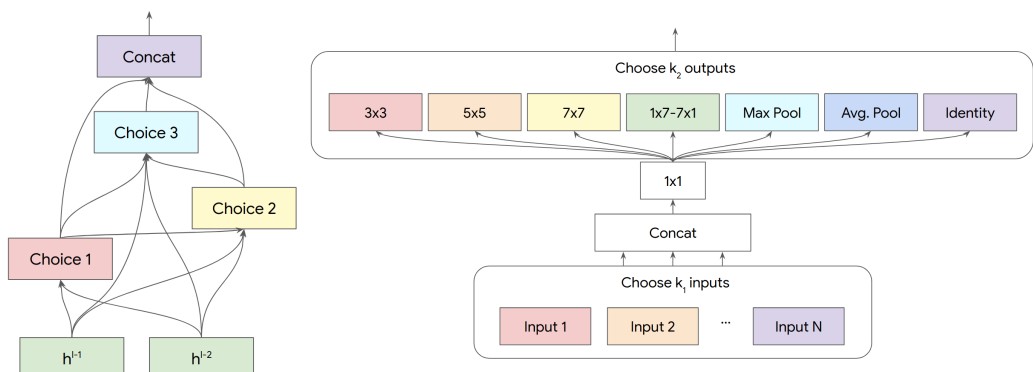

Figure 3: A neural architecture search cell of Bender et al. (2018) in the diverse routing framework. The left image depicts how the cell is constructed with choice blocks, and the right image depicts the internals of each choice block.

Thus, routing models with diversity can implement complex architectures with powerful representational capacity. We take the example of implementing a neural architecture search cell from the one-shot architecture search model of Bender et al. (2018), which is depicted in Figure 3. The model in Bender et al. (2018) consists of stacking the same cell structure multiple times. The inputs to each cell are the outputs of the previous two cells. The cell is built by stacking numerous *choice blocks*. Each choice block takes as input the cell inputs and the outputs of all previous choice blocks in the cell. Within each choice block, two decisions are made: what inputs to use, and what convolutional operations to apply. After choosing inputs, the aggregated input representation is passed to a router that selects between the different convolution, pooling, and identity operations. The input representation is fed to the selected operations, and the outputs of each operation are aggregated. This complex architecture can be readily implemented as a diverse routing model. The main differences compared to Bender et al. (2018) are that (a) in the diverse routing models, each example is routed differently, whereas Bender et al. (2018) uses a single path for all examples, and (b) for diverse routing models, each cell is different, whereas for Bender et al. (2018), a single cell architecture is chosen and replicated across the network.

## 2.3 ROUTING DEPTH

Adding diversity improves representational power in one respect, but another important dimension is routing depth. As networks have gotten deeper, their modeling performance has improved (Krizhevsky et al., 2012; Szegedy et al., 2015; Zoph et al., 2017; Mahajan et al., 2018). In order to be useful, routing models need to be able to scale to large depths. Computationally, scaling is feasible on modern hardware because routing models have block-sparse structure (Gray et al., 2017), where each expert applies dense computation but only a few experts are active. Unfortunately most routing works have been focused on small scale models with few routing decisions (Fernando et al., 2017; McGill & Perona, 2017; Rosenbaum et al., 2018).

Shazeer et al. (2017) was the first work on routing to focus on scale. They built models with hundreds of billions of parameters. While impressive, this focus on sheer number of parameters neglects the impact of routing depth. We define routing depth as the number of internal routing decisions made. In a simplified model, if there are $n$ routing decisions and $E$ experts to choose per decision, then there are $E^n$ paths in the routing model. We hypothesize that having more paths increases representational power. Shazeer et al. (2017) focused on increasing $E$, but we are interested in increasing $n$ because the path count grows exponentially in $n$.

However, as the routing depth $n$ increases, more hard decisions are made so optimizing the routing model becomes more challenging. In our experiments, we aim to benchmark the tradeoff between routing depth and ease of optimization. We focus on evaluating the noisy top-$k$ gating of Shazeer et al. (2017) because it has demonstrated strong empirical performance and can easily scale to $|E|$ in the thousands. The goal is to determine if scaling up the routing depth $n$ is feasible with modern techniques, or if additional research is required for large $n$.

## 3 RELATED WORK

Routing models fall into a category of models that utilize conditional computation (Bengio et al., 2013; Davis & Arel, 2013; Cho & Bengio, 2014), where each example only activates a portion of the entire network. Conditional computation can enable an increase in model capacity without a proportional increase in computational cost, making it an attractive method for scaling up models. A number of methods have been proposed to utilize conditional computation to speed up inference by learning when to stop early (Graves, 2016; Guan et al., 2017; McGill & Perona, 2017; Wang et al., 2017), but these methods do not learn the architectures. Our work is closest to Rosenbaum et al. (2018) that routes examples through a network in a multi-task setting. The main differences are that (a) they use experts with the same architecture, whereas we demonstrate that using diverse experts is important, and (b) they use a small routing depth while we benchmark models with large routing depth.

The routing model used in this paper builds off work from the mixture of experts field (Jacobs et al., 1991; Jordan & Jacobs, 1994; Collobert et al., 2002; Rasmussen & Ghahramani, 2002; Eigen et al., 2013). Specifically, the routers in the work utilize the noisy top-$k$ gating from Shazeer et al. (2017) to select architectural components per example. Shazeer et al. (2017) uses homogenous experts, and only make one or two routing decisions in the entire routing network. In contrast, we explore diverse routing models that make an order of magnitude more decisions. Various forms of iterative soft routing have been explored (Sabour et al., 2017), and these ideas may improve hard routers.

Diverse routing networks generalize many neural architecture search methods. Automated search techniques like neural architecture search (NAS) (Zoph & Le, 2017) have recently outperformed hand-tuned models on ImageNet (Zoph et al., 2017; Real et al., 2018). NAS style techniques have also been used to discover other neural network components, such as optimizers (Bello et al., 2017) and activation functions (Ramachandran et al., 2018). Unfortunately, architecture search tends to be extremely computationally expensive (Zoph et al., 2017; Real et al., 2018). This constraint has motivated one-shot NAS models (Pham et al., 2018; Brock et al., 2017; Bender et al., 2018) that share parameters between different candidate architectures to improve training efficiency. There have also been recent works in automatically learning and tuning architectures as part of the training procedure (Gordon et al., 2017; Huang et al., 2017). However, all these focus on static models, while we focus on routing models that differ per example.

Other categories of input-dependent models are tree-structured recursive neural networks (Socher et al., 2013) and parameter prediction models (Denil et al., 2013; Ha et al., 2016; Jia et al., 2016; Gilmer et al., 2017). Recursive neural networks match the unique tree structure of each input but do not allow for learned structural changes. Parameter prediction models infer a set of example-dependent parameters instead of using the same set of parameters for each example. However, in parameter prediction works, each example still passes through the same architecture, so the architecture remains static while the parameters differ. In diverse routing models, the architecture changes while the parameters are static. We leave it to future work to combine these two ideas to create an extremely flexible model.

## 4 EXPERIMENTS

### 4.1 EXPERT SPECIALIZATION

Before testing diversity and increasing routing depth, we verify that experts are able to capture dataset structure when trained with noisy top-$k$ gating. We train a routing model on MNIST that consists of 3 shared layers followed by an expert layer with 5 experts and $k = 2$. For simplicity, each layer and each expert is defined as a fully connected layer of 64 neurons followed by LayerNorm (Ba et al., 2016) and ReLU. After training, examples from the test set are passed through the routing model and the expert choices for each example are aggregated. The results are plotted in Figure 4.

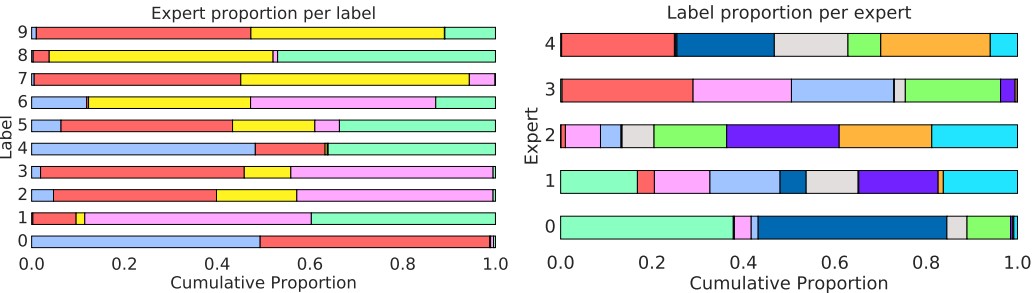

Figure 4: *(Left)* the proportion of experts assigned to examples of each MNIST class. *(Right)* the proportion of labels assigned to each expert. Different colors distinguish different experts / labels.

As shown in Figure 4 (left), examples from a particular class tend to be routed through the same pair of experts, demonstrating that experts focus on certain classes. Figure 4 (right) shows how each expert handles a different label distribution, providing additional evidence of specialization. We found normalization useful in stabilizing results across runs, so we use normalization in all following experiments.

### 4.2 DIVERSITY

Next, we benchmark routing models with architectural diversity on an Omniglot (Lake et al., 2015) multi-task learning setup. The Omniglot dataset consists of 50 distinct alphabets, and each alphabet consists of a number of characters, with 20 grayscale 105×105 handwritten examples per character. We use the multi-task setup introduced by Meyerson & Miikkulainen (2018), where each task is defined as predicting the character identity of an example within a single alphabet, and each alphabet is treated as a separate task. We follow Liang et al. (2018) by defining a 50%/20%/30% training/validation/test split and using a fixed random subset of 20 alphabets. The entire dataset consists of just 6680 training examples, making it a challenging low-resource benchmark where the danger of overfitting is especially pronounced for routing models.

The baseline is the previous state-of-the-art technique, *CMTR*, which performs automated architecture search on multiple tasks using hierarchical evolutionary algorithms (Liang et al., 2018). A big limitation of their method is that the training time scales linearly with the number of tasks. The aforementioned 20 task subset is because CMTR is too computationally expensive to perform on 50 tasks. In contrast, the diverse routing model can easily scale to many tasks.

The routing model used is a convolutional neural network with 1 shared layer and 8 expert layers, each with 48 filters. Each task is assigned a separate linear readout layer and task-specific embeddings are additionally fed to each routing model. Each expert layer consists of 7 different types of experts: conv $3 \times 3 \to$ conv $3 \times 3$, conv $5 \times 5 \to$ conv $5 \times 5$, conv $7 \times 7 \to$ conv $7 \times 7$, conv $1 \times 7 \to$ conv $7 \times 1$, $3 \times 3$ max pooling, $3 \times 3$ average pooling, and an identity layer. Each convolution and expert layer is followed by GroupNorm (Wu & He, 2018) and ReLU. $k$ is annealed from 7 to 2 over the layers. We found the $k$-annealing technique crucial to prevent overfitting. The Adam optimizer (Kingma & Ba, 2014) is used, and the expert-balancing for noisy top-$k$ loss is annealed from 0.1 to 0 over the course of training.

Table 1: Results on Omniglot multi-task learning. Results are taken over 10 runs. The first two results are from Meyerson & Miikkulainen (2018) and the next four results are from Liang et al. (2018).

| Algorithm | Valid Error (%) | Test Error (%) |
|---|---|---|
| Single Task | $36.41 \pm 0.53$ | $39.19 \pm 0.50$ |
| Soft Ordering | $32.33 \pm 0.74$ | $33.41 \pm 0.71$ |
| CM | $19.62 \pm 0.36$ | $18.67 \pm 0.27$ |
| CMSR | $16.31 \pm 0.21$ | $16.18 \pm 0.18$ |
| CMR | $17.52 \pm 0.21$ | $17.64 \pm 0.19$ |
| CMTR | $11.80 \pm 1.02$ | $12.18 \pm 1.02$ |
| Routing (ours) | $\mathbf{7.95 \pm 0.37}$ | $\mathbf{7.81 \pm 0.54}$ |

Table 1 shows how a diverse routing model achieves more than a 35% error reduction compared to the previous state-of-the-art. The best models of Liang et al. (2018) have around 3M parameters, while our routing models have 1.8M parameters. The CMTR method has the ability to modify parameter count, so the routing model is not better due to parameter count differences. These results demonstrate that routing models, if properly regularized (such as with the $k$-anneal trick), can suceed in low-resource tasks. Furthermore, the model takes less than one hour to train on a single GPU. which is especially notable given that it outperforms CMTR, which is a computationally expensive architecture search method that directly optimizes validation accuracy. This result suggests that the diverse routing models are effectively performing a form of architecture search.

### 4.2.1 OMNIGLOT ABLATIONS

It is unclear whether the architectural diversity of the experts is important for achieving good results. If diversity is important, then removing diversity while maintaining model size should reduce accuracy. To clear this doubt, we run an ablation study on Omniglot by changing the expert architecture in the routing model described above. Instead of using multiple types of convolution, pooling, and identity layers, we try three different sets of experts: (a) each expert is a conv $3 \times 3 \to$ conv $3 \times 3$, (b) each expert is a conv $5 \times 5 \to$ conv $5 \times 5$, and (c) each expert is one of the four convolutions in the standard architecture (i.e., the pooling and identity experts are replaced with convolutions).[1] All other architectural details stay the same, including using the same number of experts. Hyperparameters such as learning rate are retuned separately for each ablation.

Table 2: Ablation study over the importance of diversity in expert architectures.

| Expert types | Valid Error (%) | Test Error (%) |
|---|---|---|
| (a) Only conv $3 \times 3$ | $15.80 \pm 0.93$ | $16.65 \pm 0.81$ |
| (b) Only conv $5 \times 5$ | $12.10 \pm 1.00$ | $11.96 \pm 0.92$ |
| (c) Only convs | $10.52 \pm 0.78$ | $10.67 \pm 0.80$ |
| Conv, pooling, identity | $\mathbf{7.95 \pm 0.37}$ | $\mathbf{7.81 \pm 0.54}$ |

Table 2 shows the results of the ablation. Routing models with less diversity perform worse. Using the same type of convolution performs poorly, and even using a diverse set of convolutions performs

---

[1]Specifically, there are two instances of conv $3 \times 3 \to$ conv $3 \times 3$, conv $5 \times 5 \to$ conv $5 \times 5$, and conv $7 \times 7 \to$ conv $7 \times 7$ each, and one instance of conv $1 \times 7 \to$ conv $7 \times 1$,

worse than using convolutions, pooling, and identity operations. Since it is challenging to know apriori what operations are important for a given task, using diverse architectural components simplifies modeling and improves performance in routing models.

### 4.2.2 Additional Ablations Across Datasets

Table 3: Accuracy of ResNet-50 style routing models with only 3×3 experts or all expert types across 4 datasets. For fair comparison, all models have been hyperparameter tuned equally and independently. Bold indicates a nontrivial difference that cannot be explained by noise.

|  | Stanford Cars | Sun397 | Birdsnap | Food101 |
|---|---|---|---|---|
| # train examples | 8144 | 19850 | 47386 | 75750 |
| Only conv $3 \times 3$ | 87.8 | **47.2** | 68.8 | 85.0 |
| All diverse experts | **89.1** | 45.2 | 68.5 | 85.0 |

We additionally compare homogeneous and diverse models across multiple image datasets: Stanford Cars (Krause et al., 2013), Sun397 (Xiao et al., 2010), Birdsnap (Berg et al., 2014), and Food101 (Bossard et al., 2014). The routing model used is a modified ResNet-50 (He et al., 2016). All non-downsampling bottleneck conv $3 \times 3$ layers were replaced by a routing layer. We use 5 types of experts: conv $3 \times 3$ , conv $5 \times 5$, conv $1 \times 7 \to$ conv $7 \times 1$, $3 \times 3$ max pooling, and squeeze-and-excitation (Hu et al., 2017). We compared this all-experts model to a homogeneous routing model with 5 conv $3 \times 3$ experts per routing layer. The parameter count for both models are close: 50M for the homogeneous model and 52M for the all-expert model. We follow the settings of Kornblith et al. (2018), but double the number of training steps which we found improves the results of both models. Each model is independently hyperparameter tuned with a grid search over 4 possible learning rates and 4 possible weight decays on a held out validation set. A final model with the best hyperparameter setting is trained on the whole training set and then evaluated on the test set once.

Table 3 shows the results of the ablation. The performance of diverse models compared with homogeneous models is dataset dependent, without a strong correlation to dataset size. The dependence of ranking on dataset is not surprising, as the ranks of standard models are typically also dataset dependent (Kornblith et al., 2018). In fact, since it is challenging to know apriori which set of operations work well for which dataset, this result suggests that adding diversity of operations to models in addition to adding multiple copies of a single important operation type (such as conv 3×3) is a straightforward way of creating routing models that work well for different datasets without requiring much tuning. This suggestion connects back to the architecture search view of diverse routing models, where diverse routing models generalize standard architecture search methods and pick the best operations to use. Furthermore, since the number of experts used per example is independent of the total number of experts, the running time is unaffected. Thus, we suggest that future routing works combine operation diversity with operation cloning.

### 4.3 Routing Depth

Finally, we attempt to scale up the routing depth. Prior work on routing has employed models with a routing depth of 5 or less. Our aim is to benchmark how successful noisy top-$k$ gating performs when tasked with making a large number of decisions. In this experiment, the routing models perform 48 or 96 decisions, which is an order of magnitude larger than prior works.

We evaluate the routing models on the standard CIFAR-10 dataset. In order to ensure that routing decisions play a nontrivial role in the performance of the model, we leverage the neural architecture search cell of Bender et al. (2018) whose set of operations has been tuned to maximize performance while removing unnecessary operations. We follow the implementation as detailed in Section 2.2, and experiment with 6-cell and 12-cell models. The 6-cell model has 48 internal decisions while the 12-cell model has 96 internal decisions. The $k$-annealing trick is used in all routing models.

The results are shown in Table 4. The routing models perform competitively with the one-shot and all-on models. However, when evaluated in terms of parameters, routing models perform worse. For example, the routing model with $C = 6, F = 64$ performs as well as an all-on model with $2\times$ fewer parameters ($C = 6, F = 32$) and a one-shot model with $3.5\times$ fewer parameters ($C = 6, F = 32$).

Table 4: Results on CIFAR-10. $F$ is the base number of filters in each cell, and $C$ is the number of cells. The *one-shot* and *all on* come from Bender et al. (2018). The all-on model is one where all operations are used for every example. The one-shot top model is a retrained static architecture which is a subset of the all-on model.

| Method | Avg. params / example (M) | Accuracy (%) |
|---|---|---|
| One-shot Top ($C = 6, F = 16$) | 0.7 | 94.6 |
| One-shot Top ($C = 6, F = 32$) | 2.7 | 95.5 |
| One-shot Top ($C = 6, F = 64$) | 10.4 | 95.9 |
| One-shot Top ($C = 6, F = 128$) | 41.3 | 96.1 |
| All-On ($C = 6, F = 16$) | 1.3 | 95.0 |
| All-On ($C = 6, F = 32$) | 4.8 | 95.6 |
| All-On ($C = 6, F = 64$) | 18.5 | 96.0 |
| All-On ($C = 6, F = 128$) | 72.7 | 96.2 |
| Routing (ours) ($C = 6, F = 32$) | 2.5 | 95.1 |
| Routing (ours) ($C = 6, F = 64$) | 9.6 | 95.5 |
| Routing (ours) ($C = 12, F = 32$) | 7.8 | 94.7 |

Since the representational capacity of the routing model is no worse than the models from Bender et al. (2018), we suspect that the complicated optimization of routing models hurts performance. Additional evidence in support of this argument is the fact that the 12-cell model slightly underperforms both 6-cell models. The 12-cell model has $2\times$ the number of decision nodes, making it an even more challenging optimization problem. Thus, this negative result demonstrates that while increasing routing depth should theoretically improve routing models, current techniques for optimizing the router, such as noisy top-$k$ gating, are empirically unable to learn strong solutions.

## 5 DISCUSSION

In this work, we introduced diversity to routing models and experimented with increasing routing depth. We believe that these two ideas are both intuitive and simple for researchers to implement. In our experiments, we found that architectural diversity can have a big impact in final performance. However, the impact of routing depth remains uncertain due to optimization difficulties faced by current methods.

While routing models are a promising direction of research, practitioners still prefer static models due to their simplicity and reliable performance. For the use of routing models to become widespread, there must be a successful application of routing models on a domain where static models struggle. We believe that large scale problems fit this criteria, since they play into the theoretical scaling strengths of routing models. While architectural diversity will help improve routing models on large scale tasks, the routing depth optimization problem will continue to impede success. We encourage researchers to develop methods that will enable routing methods to effectively scale on large scale tasks. We remain optimistic that routing models will play an important role in the neural networks of the future.

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
