# OpenReview forum: "Diversity and Depth in Per-Example Routing Models"
_ICLR.cc/2019/Conference_

### Official Review · AnonReviewer3 · 2018-11-02
**Very interesting discourse, but experiments could be more thorough.**

**Rating:** 6
**Confidence:** 5

**Review:**

The major contribution of this work is extending routing networks (Rosenbaum et al., ICLR 2018) to use diverse architectures across routed modules. I view this as an important contribution and am very impressed by the experiment on Omniglot where it shows big performance gain on a split with very few examples. This idea of incorporating in architectural bias and not just parameter bias for small data problems is very compelling and intuitive to me on the surface. The ablation study was also very interesting in this regard. I really like the discourse and found it to be filled with interesting insights throughout ranging from the connection between routing networks and neural architecture search to the heuristic for selecting k.  However, after the great discourse, I was quite disappointed by the breadth of the experiments.

The paper is positioned as exploring two parallel ideas that are independently interesting 1) diversity in the architecture of modules in routing models 2) the effect of increasing depth in routing models. For the first idea, this is shown very well by the Omniglot experiment but is not evaluated in any other setting. Showing this in a few other experiments would have really driven this point home in my opinion.  The second idea is not really executed in a convincing way to me. The authors call it a ‘negative result’ in the end, but I’m not sure I really feel like I learned anything from this experiment. I wonder about statistical significance. I also feel like the authors are trying to turn it into a commentary that this is a pain point for all variants of routing models while they only actually tried it for their proposed architecture which makes quite a few decisions along the way. I would have liked to see more model variants and datasets before really feeling like I can make any empirical determinations about the fundamental limitations of all routing models in this regard.  Additionally, if there were such a fundamental scaling limitation, you would imagine that an experiment could be constructed that really highlighted this fact where all routing models do way worse.

In short, I think there are some really good idea in this paper and vote for acceptance on that basis. Had the authors provided more empirical evidence about architectural diversity, I would have given it a very high score. The analysis of depth is also a very interesting topic, but it could possibly even serve as another paper considering that the current results don’t really come to concrete conclusions for the community.

---

> ### Author Response · Authors · 2018-11-27
> **Re: Reviewer 3**
>
> Thank you for the comments.
>
>
> >>> re: additional experiments about architectural diversity
>
> We have run experiments with a different style of model based on ResNet on an additional 4 datasets with an eye towards fair parameter count and hyperparameter tuning. Please see Section 4.2.2 for more details and analysis.
>
>
> >>> re: additional experiments about depth
>
> We are currently in the middle of running additional experiments comparing depth, and will update the paper once these experiments complete. The setup compares ResNet-50 and ResNet-101 models with routing. Preliminary results in this setup also suggest that increased routing depth makes models harder to train. For example, on the Food101 dataset with 75K examples, the ResNet-101 routing model achieves around a 5% worse accuracy than the ResNet-50 routing model (both models have been hyperparameter tuned the same amount). This result cannot be explained by overfitting, because Food101 has more examples than CIFAR-10, where larger models with sufficient regularization outperform smaller models. We use aggressive Imagenet-style data augmentation which should provide sufficient regularization. The accuracy drop of deeper routing models is so far consistent across all the datasets we have evaluated.
>
>
> >>> re: “the fundamental limitations of all routing models in this regard”
>
> We want to clarify that we do not believe routing models are fundamentally limited with respect to depth. Our assertion is that current routing methods do not perform well when depth is scaled up. In order for routing models to succeed, this flaw must be fixed. We draw an analogy to sequence models such as RNNs. For many years, training sequence models with long sequence lengths was impossible. However, as a consequence of numerous recent advances, sequence models can now scale to impressive sequence lengths (e.g., [1] successfully trains with a sequence length of 11000). In the same vein, we hope our analysis of depth in our work will spur discoveries of new routing techniques that will overcome the depth scaling problem.
>
> [1] Liu, Peter J, Saleh, Mohammad, Pot, Etienne, Goodrich, Ben, Sepassi, Ryan, Kaiser, Lukasz & Shazeer, Noam. “Generating wikipedia by summarizing long sequences”. ICLR 2018.

---

### Official Review · AnonReviewer1 · 2018-11-05
**Diversity and Depth in Per-Example Routing Models - Review**

**Rating:** 6
**Confidence:** 4

**Review:**

Overall, this is a valuable read. Authors tackle the head on problem of what is a good architecture where we can having routing with diverse models. The papers is written well, with comparisons to mixture of experts, other models that tackled this problem with either homogeneous architectures or static architectures. Below is my assessment on various axis:

Quality - Enough experiments to justify some conclusions, equations helped ground the method with math.

clarity - Very well written, good figures and analysis.

originality - While the authors achieve SOA results on OmniGlot and do explore a few options, I feel the work still lacks originality in the formulation or does not have original contributions to either the architectures used or the optimization procedures employed.

significance - very significant to look at this problem both in terms of compute, accuracy perspective as well as scaling these networks for multiple tasks.

pros - thorough analysis, even the negative experiments are well written and throw more light into the problem space.

cons - OmniGlot comparisons seem not fair as the model capacity is not added as part of the table which raises concerns on achieving state of the art with more high complexity models than routing mechanism. Will be great to move from CIFAR-10 and test things on CIFAR-100 to really see the value of proposed work. I would recommend a higher rating if authors address these two concerns.

---

> ### Author Response · Authors · 2018-11-27
> **Re: Reviewer 1**
>
> Thank you for the comments.
>
>
> >>> re: “OmniGlot comparisons seem not fair as the model capacity is not added as part of the table which raises concerns on achieving state of the art with more high complexity models than routing mechanism.”
>
> Liang et al. (2018), who holds the previous state-of-the-art, state that their best models have 3M parameters. Our routing model has 1.8M parameters. Thus, a higher parameter count cannot explain the difference. Furthermore, the architecture search method of Liang et al. (2018) has the ability to significantly modify the parameter count, so a lower parameter count also cannot explain the difference since small models were available in the search space. We have clarified this point in the text.
>
>
> >>> re: additional experiments on more datasets
>
> We have run experiments with a different style of model based on ResNet on an additional 4 datasets with an eye towards fair parameter count and hyperparameter tuning. Please see Section 4.2.2 for more details and analysis.

---

### Official Review · AnonReviewer4 · 2018-11-12
**Good paper with insufficient experiments**

**Rating:** 7
**Confidence:** 5

**Review:**

The paper "Diversity and Depth in Per-Example Routing Models" extends previous work on routing networks by adding diversity to the type of architectural unit available for the router at each decision and by scaling to deeper networks. They evaluate their approach on Omniglot, where they achieve state of the art performance.

Overall, the paper is very well written and every aspect can be easily understood. The overview over related work given in the paper is thorough, and the authors explain very well how their approach relates to previous approaches.

The architecture presented is a natural and important extension of previous work. Adding diversity in routing units has indeed not been investigated well and is an important contribution to the community. Additionally, the authors do a good job of identifying problems with existing approaches (overfitting, routing depth) and offer a empirically convincing solutions.

The result section given in the paper is its weakness and requires a more in-depth analysis:
+ the results given for Omniglot are impressive
+ the experiments analyzing the impact of diversity and routing depth are interesting and offer interesting insight into the architecture
- the results do not show learning behavior over epochs; this is not necessary, but would give an additional insight into the learning behavior of the architecture
- the experimental settings are confusing: why are the different experiments performed with different datasets? This makes it seem as if the authors cherry-picked the best results for the different experiments (this might not be the case, but the results on Omniglot alone are good enough that negative results and a detailed discussion of them would not have hurt the paper, but enriched the discourse)
- additional experiments that offer a transition from larger datasets to smaller ones would be interesting; seeing how the performance of the architecture behaves e.g. on CIFAR10 for 1k, 5k, 10k, 25k and 50k would have illustrated how well the architecture is able to generalize from different numbers of samples

In summary, I think the paper analyzes a very important problem and has a lot of potential. However, it needs more extensive experiments that illustrate how the proposed architecture behaves over a wider variety of datasets.

UPDATE AFTER REBUTTAL:
I am still torn about this paper. On one hand, I still think that the topic and discourse provided by this paper is extremely important. On the other, the results - even after the revision - do not completely convince me. I might update my score after some discussion with the other reviewers.
2ND UPDATE:
After giving it some more thought, I find myself convinced that this paper has a contribution important enough to be accepted. I increase my score to 7.

---

> ### Author Response · Authors · 2018-11-27
> **Re: Reviewer 4**
>
> Thank you for the comments.
>
>
> >>> re: learning behavior over epochs
>
> We have found that the training behavior is surprisingly unremarkable. We had initially hypothesized that there would nontrivial steps in the loss curve when the model learns to correctly route a whole class of examples, but in practice, the loss curve is smooth. This finding is especially interesting given that the routing depth experiments demonstrate that there are optimization difficulties. One interpretation may be that the loss landscape is smooth but increasing the routing depth adds many more suboptimal minima / saddle points that the optimization procedure can get stuck in.
>
>
> >>> re: “why are the different experiments performed with different datasets”
>
> We want to clarify that we did not cherrypick experiments. Our very first experiments on diversity were on Omniglot due to the strong architecture search baseline of Liang et al. (2018). We then chose CIFAR-10 for the depth experiments because it is a standard dataset where the community has determined that increased depth and model size consistently improves results. Thus, we can clearly identify if optimization is problematic. It’s unclear if this property holds for Omniglot since overfitting of large models may play a factor in poor performance, making disentangling the effects of optimization difficult.
>
>
> >>> re: additional experiments that offer a transition from larger datasets to smaller ones
>
> We have run experiments with a different style of model based on ResNet on an additional 4 datasets with an eye towards fair parameter count and hyperparameter tuning. The 4 datasets span a range of sizes. Please see Section 4.2.2 for more details and analysis.

---

> > ### Comment · AnonReviewer4 · 2018-11-30
> > **Parameter Count and Omniglot**
> >
> > @Reviewer 1: I disagree with the statement that the omniglot results do not seem fair because of parameter size. The point here is that parameter count is not particularly important for a problem with as few training samples as Omniglot. On the contrary - large models can overfit more easily and consequently generalize less. However, I would be interested to get the authors's sense of why this does not happen here ...?
> >
> > @Reviewer 3: Unfortunately, I very much share Reviewer 3's sentiments. If the paper contained a more thorough experimental analysis on a wider range of datasets, I think that a very high acceptance score would have been adequate. Without, I am not sure as to how the paper scales to different problems.
> >
> > ( I originally thought I could comment on other reviews directly, hence this format. I apologize if it causes any confusion.)

---

> > > ### Author Response · Authors · 2018-12-03
> > > **Re: Parameter Count and Omniglot**
> > >
> > > First of all, thank you for following up!
> > >
> > >
> > > >>> re: overfitting
> > >
> > > Indeed, we are also quite surprised that these models can perform so well on low resource tasks. However, this phenomena is not unique to our model and actually reflects an observation made by numerous researchers in our community. Namely, massively overparameterized models seem to perform better than models with fewer parameters, even if the number of examples in the dataset is orders of magnitude smaller than parameters. The most recent example of this phenomena is Huang et al. (2018) [1], who train a model with 557M parameters and achieve a new SOTA (for randomly initialized models) on Imagenet, which has 1.28M images. When they finetune on CIFAR-10, which has 50K images, they also set a new SOTA at 1.0% error rate. Analyzing this phenomena is an active area of research right now (for example, see Arora et al. (2018) [2]), and we too are looking forward to a better understanding of this behavior.
> > >
> > >
> > > >>> re: more datasets
> > >
> > > We added a new revision during the revision period that analyzed more 4 more datasets and a new type of model. We focused on providing fair comparisons between models (e.g., matching parameter count, hyperparameter tuning models the same amount). Please take a look at Section 4.2.2 in our updated paper.
> > >
> > > As a more general point, we believe that routing models are still in the "ugly duckling" phase, where there are pockets of interesting results, but nothing yet has been truly convincing. We draw an analogy to deep learning before Alexnet. There were some interesting results (such as in phoneme recognition), but most researchers in the community at that time did not anticipate that deep learning would change the field so much. It took a culmination of many different threads of research to hit the breakthrough: larger datasets, better computation, and a series of small but important changes to neural networks, such as ReLU and better initialization.
> > >
> > > Routing networks are still in the phase before the threads come together. For example, one current challenge is finding a problem setting where routing models have a distinct advantage over standard neural networks. This best problem setting may be a low latency embedded setting [3], or it may be a setting where one wants to train the largest model possible [4]. Different researchers have been exploring these various problem settings (the analogy is constructing a large dataset, Imagenet, which let neural networks shine). Another important but orthogonal direction is improving routing models themselves (the analogy is ReLU and better initialization). Our work falls in this area of research, where we explore diversity and also question if current techniques are sufficient for achieving scale. The third line of research, analogous to computation, is the recent push in machine learning systems like Tensorflow for better support of model parallelism and sparse computation in general. Thus, despite there being a lack of game-changing results achieved by routing models, we believe that if the community continues to push along these directions, the threads may come together and we may find a big success down the line.
> > >
> > >
> > > References
> > > ------------------
> > >
> > > [1]  Huang, Y., Cheng, Y., Chen, D., Lee, H., Ngiam, J., Le, Q. V., & Chen, Z. (2018). GPipe: Efficient Training of Giant Neural Networks using Pipeline Parallelism. arXiv preprint arXiv:1811.06965.
> > > [2] Arora, S., Cohen, N., & Hazan, E. (2018). On the optimization of deep networks: Implicit acceleration by overparameterization. arXiv preprint arXiv:1802.06509.
> > > [3] Teerapittayanon, S., McDanel, B., & Kung, H. T. (2016, December). Branchynet: Fast inference via early exiting from deep neural networks. In Pattern Recognition (ICPR), 2016 23rd International Conference on (pp. 2464-2469). IEEE.
> > > [4] Shazeer, N., Mirhoseini, A., Maziarz, K., Davis, A., Le, Q., Hinton, G., & Dean, J. (2017). Outrageously large neural networks: The sparsely-gated mixture-of-experts layer. arXiv preprint arXiv:1701.06538.

---

### Meta-Review · Area_Chair1 · 2018-12-13
**Architectural diversity helps routing**

**Confidence:** 5
**Recommendation:** Accept (Poster)

**Metareview:**


pros:
- good, clear writing
- interesting analysis
- very important research area
- nice results on multi-task omniglot

cons:
- somewhat limited experimental evaluation

The reviewers I think all agree that the work is interesting and the paper well-written. I think there is still a need for more thorough experiments (which it sounds like the authors are undertaking).  I recommend acceptance.